# Continental Thunderstorm Ground Enhancement observed at an exceptionally low altitude

Ivana Kolmašová[1,2], Ondřej Santolík[1,2], Jakub Šlegl[3,4], Jana Popová[1], Zbyněk Sokol[1], Petr Zacharov[1], Ondřej Ploc[3], Gerhard Diendorfer[5], Ronald Langer[6,3], Radek Lán[1], and Igor Strhárský[6]

[1]Institute of Atmospheric Physics of the Czech Academy of Sciences, Prague, Czechia
[2] Faculty of Mathematics and Physics, Charles University, Prague, Czechia
[3] Nuclear Physics Institute of the Czech Academy of Sciences, Husinec-Rez, Czechia
[4] Faculty of Nuclear Sciences and Physical Engineering, Czech Technical University in Prague, Prague, Czechia
[5] Department of ALDIS, OVE Service GmbH, Vienna, Austria
[6] Institute of Experimental Physics, Slovak Academy of Sciences, Kosice, Slovakia

*Correspondence to*: Ivana Kolmašová (iko@ufa.cas.cz)

**Abstract.** Two long-lasting Thunderstorm Ground Enhancement (TGE) events were registered at the Milešovka meteorological observatory in Czechia (50.55N, 13.93E, altitude 837 m) on 23 April 2018, during linearly organized thunderstorms. Two intervals of increased photon counts were detected by a plastic scintillator, respectively lasting 70 and 25 minutes, and reaching 31% and 48% above the background radiation levels. Using numerical simulations, we verified that the observed increases of count rates are consistent with the energy spectrum of previously observed TGEs. We investigated the relevant data from a suite of meteorological instruments, a Ka-band cloud radar, an electric field mill, and a broadband electromagnetic receiver, all placed at the Milešovka observatory, in order to analyze the context in which these unique continental TGEs occurred at an exceptionally low altitude. The onset of the TGEs preceded the onset of precipitation by 10 and 3 minutes, respectively, for the two events. Both this delayed rain arrival and a lower energy threshold of 6.5 MeV for registered particles clearly exclude the detection the decay products of the radon progeny washout during the TGE intervals. At the same time, the European lightning detection network EUCLID detected numerous predominantly negative intracloud lightning discharges at distances closer than 5 km from the particle detector, while the occurrence of cloud-to-ground discharges was suppressed. The cloud radar recorded presence of graupel below the melting level and the composition of hydrometeors suggested good conditions for cloud electrification. The observed variations of the near surface electric field were unusual, with very brief negative electric field excursions reaching -20 kV in a quick succession. At the same time, sub-microsecond unipolar pulses emitted by close corona discharges saturated the broadband magnetic loop antenna. All these measurements indicate that a strong lower positive charge region was present inside the thundercloud. The lower thundercloud dipole was probably responsible for acceleration of the seed electrons in the air. These seed electrons might originate not only in the secondary cosmic ray particles but could also come from a high concentration of radon in the air collected during the propagation of the convective system above the uranium-rich soils before the thunderstorms overpassed the Milešovka observatory.

## 1 Introduction

"Thunderstorm Ground Enhancement" events are defined as increased fluxes of electrons, neutrons, gamma rays or X rays, which are registered by particle detectors located on the Earth's surface during thunderstorms (Chilingarian et al., 2010, 2011, 2015, 2016, 2019; Kudela et al., 2017; Chum et al. 2020). These phenomena are also known as "prolonged radiation bursts" (Tsuchiya et al., 2007), "gamma glows from the ground" (Dwyer et al., 2012), "prolonged gamma ray enhancements" (Shepetov et al., 2021), or "gamma ray bursts of atmospheric origin" (Brunetti et al., 2000). The first conclusive measurements of these "X-ray enhancements" clearly related to thunderstorms were obtained using airplanes (Parks et al., 1981), followed by "X-ray increases" on balloons (Eack et al., 1996), and by airborne measurements of "gamma ray glows" (Kelley et al., 2015; Kochkin, 2017; Ostgaard, 2019). However, the first theoretical prediction of "extremely penetrating radiation of beta or gamma ray type" was published by Wilson (1925) who hypothesized that beta radiation might come from energetic electrons, accelerated by thunderstorm electric fields from the seed population of decay products of cosmic rays or radionuclides of terrestrial origin, while the gamma component might come from bremsstrahlung after collisions of these electrons with the air molecules. Given the above documented fact, that most papers on this subject coin their proper term to name these interesting phenomena, we have a wide choice of possible names of which we chose the term "Thunderstorm Ground Enhancement (TGE)" which currently occurs most often in the literature.

The main complications for observations of TGEs were (a) emissions originating in the decay chain of the radon (mostly 214Br and 214Pb) washed out from the air by rain and (b) highly absorbing column of the air between the cloud base and the detector (Dwyer et al., 2012). The origin of radon and its progeny in the air was explained (Chilingarian et al.,2020a) by their attaching to charged aerosols after being lifted by the near surface electric field to the air. Their radiation then can be registered by particle detectors simultaneously with the TGE particles. Rain quickly returns some of the isotopes back to the ground. In the absence of rain, the radiation from the air can continue for 1-2 hours until radon progeny finally decays. The exclusion of the radon progeny washout and its subsequent decay products (at energies below 3 MeV) in the registered counts started to be possible with an extension of measured particle energies up to 10 MeV. The absorption can be minimized by choosing observational places with a short distance between the cloud base and the detector. This is the reason, why the TGEs were up to now exclusively observed at high mountain observatories (Brunetti et al., 2000; Torii et al, 2009; Chilingarian et al., 2011, 2015, 2016; Kudela et al., 2016; Chum et al. 2020, Shepetov et al., 2021) or at the sea level during Japanese winter storms with extremely low cloud base altitudes (Tsuchiya et al., 2011; Kuroda et al., 2016). Typically, the TGEs last from 1 minute up to 10-15 minutes and the radiation mostly does not exceed 10 % of the background values. Nevertheless, extreme events exceeding several times the background values were also registered (Chilingarian et al., 2010; Chum et al., 2020).

Chilingarian et al. (2012) introduced a two-component model for the TGE generation, which includes the Relativistic Runaway Electron Avalanche (RREA) process originally proposed by Gurevich et al. (1992) for the thunderstorm electric fields above

70 the RREA threshold, together with the Modification of electron energy Spectra (MOS) process for high-energy electrons and for electric fields both below and above the RREA threshold.. RREA might be responsible for multiplication of particle flux up to 10 times above the background of secondary cosmic rays in the energy range up to 30-40 MeV. The MOS process can add only several per cent particle flux to the background values but the energy extends up to 100 MeV. Dwyer and Uman (2014) showed that an avalanche could be produced in the thundercloud electric fields if energetic seed electrons are provided,

for example, by secondary cosmic rays. Energetic runaway electrons then generate high-energy photons through the bremsstrahlung interactions with air atoms. These high-energy photons can reach energies of a few tens MeV. A transfer of energy of the thundercloud electric field to the electrons from the ambient population of the cosmic rays leads to a modification of electron energy spectra, to an additional bremsstrahlung, and might be also responsible for the tail of the TGE gamma ray spectra up to 100 MeV (Chilingarian et al., 2012). Using the observed enhancements of photon and electron fluxes measured

by the upper scintillator of SEVAN at Lomnicky stit (altitude 2 634 m) and their comparison with the simulations of the RREA Chilingarian et al. (2021) showed, that the potential difference present in in the thunderous atmosphere might reach approximately 500 MV.

It was shown by simultaneous measurements of particle fluxes and near surface electric fields that TGEs usually occurred during large values of negative electric fields, which accelerate electrons downwards. Nevertheless, TGEs were occasionally

detected also during positive electric fields (Zhou et al., 2016; Kudela et al., 2017; Bartoli et al., 2018, Chum et al., 2020). TGEs are usually not associated with individual lightning strokes, but quite often they are reduced or terminated abruptly by a nearby lightning discharge (Kudela et al., 2017; Chilingarian et al., 2017a; Chum et al., 2020, Soghomonyan et al., 2021; Kochkin et al., 2021). TGEs are often observed during time intervals with an increased occurrence of inverted intracloud lightning, which are discharges between the main negative charge region and the lower positive charge region (LPCR), and

during a lower occurrence of negative cloud-to-ground (CG) lightning strokes (Chilingarian et al., 2018, 2020). This scenario suggests an existence of a strong LPCR inside the thundercloud, which blocks the propagation of negative leaders down to the ground (Nag and Rakov, 2009; Iudin et al., 2017). This arrangement of charges inside the thundercloud also suggests that electrons from the cosmic ray secondaries are accelerated and multiplied in the bottom thundercloud dipole, which is formed by the main negative charge layer and the LPCR (Chilingarian et al., 2017b). This led to a speculation that the intensity of

TGEs reached the maximum when the LPCR was directly above the detector and the counts decreased when the cloud moved away. Such movement of clouds would explain a large variety in durations and intensities of the observed TGEs. This effect was reported by Torii et al. (2011), who identified a migrating source of high-energy photons attributed to the thundercloud movement using simultaneous registrations of TGEs, measurements of the near surface atmospheric electric field, and meteorological radar echoes at several points along the Japanese coast.

The mechanism of the LPCR formation is still not fully understood. It is typically located just below the freezing level. Rakov and Uman (2003) proposed several hypothetical sources of positive charge, which can contribute to its accumulation close to the lower cloud boundary. The source of positive charge might be associated with graupels, which are supposed to be positively

charged at temperatures warmer than the reversal temperature. Valuable contribution to the LPCR puzzle can be added by information about the thundercloud microphysical structure: a mixture of hydrometeors such as that of graupel, ice, snow and supercooled water is considered prone to cloud electrification (Rakov, 2016). Such data can be delivered by millimeter Doppler polarimetric radars, which investigate the cloud microphysics at high temporal and spatial resolutions (Görsdorf et al., 2015; Kollias et al., 2007; Clothiaux et al., 1995). Positive charge might be also generated by corona discharges at ground level and transferred to an altitude of the cloud base (Chauzy and Soula, 1999). This corona mechanism was also assumed to act as the main contributor to the evolution of the LPCR in the study of Nag and Rakov (2008) who evaluated the role of the LPCR in facilitating different types of lightning. Electromagnetic pulses emitted by corona discharges might be identified in fast electromagnetic recordings from their microsecond durations, unipolarity, and random distributions (Arcanjo et al., 2021). Unipolar microsecond scale pulses were found to accompany in-cloud processes as dart leaders or K-changes, but these appeared in several hundred microsecond long pulse trains with regular inter-pulse intervals (Rakov et al., 1992; Kolmašová and Santolík, 2013). Therefore, these pulses can be distinguished from the characteristic radiation from local corona discharges observed in electromagnetic recordings. Arcanjo et al. (2021) found that corona current pulses measured at a shunt resistor have fast rise times (tens of nanoseconds) and slow decays (hundreds of nanoseconds). They also found that the pulse cadence was correlated with the ambient electric field measured at a distance of 250 m. Pulses related to positive corona discharges were reported to be no longer observed for weaker ambient electric fields than -1.8 kV/m. A threshold for negative corona pulses was higher, reaching about 3.8 kV/m.

The first attempt to examine enhancements of gamma ray background, previously attributed solely to radon progeny, was reported at the territory of Czechia by Šlegl et al. (2019). The authors used the data from the Czech Radiation Monitoring Network (RMN), which is operated by the State Office for Nuclear Safety, and investigated gamma background enhancements with respect to the proximity of thunderstorms. They found that increased exposure levels at individual RMN stations observed during close thunderstorms couldn't be explained by the radon progeny itself and suggested that they might have been attributed also to TGEs.

In the present study, we investigate conditions, which led to the observation of two TGE events detected by a particle detector at the Milešovka meteorological observatory (Czechia, 837 m a.s.l) on 23 April 2018, using the data collected by a set of instruments: an electric field mill, a broadband electromagnetic receiver, and a Ka-band cloud radar. We combine these measurements with meteorological data (temperature, precipitation, air pressure, dew point temperature), and with data provided by the European lightning location network EUCLID. In section 2, we describe the instrumental setup and the dataset. In section 3, we describe the meteorological situation during the thunderstorms occurring on 23 April 2018. In section 4, we present results of our analysis of the particle registrations. In section 5, we analyze electrostatic and electromagnetic measurements and investigate characteristics of lightning detected by EUCLID during the analyzed thunderstorms. In section

6, we introduce the relevant observations of the cloud radar. In section 7, we describe the simulation of observed particle fluxes. In section 8, we discuss and summarize our results.

## 2 Instrumentation and dataset

The Milešovka meteorological observatory is located on the top of the Milešovka Mt. (a. k. a. Donnersberg, 50.55N, 13.93E, 837 m a. s. l.) in Czechia. As it is by 400 meters higher than the surrounding terrain, and has a 360° view unobstructed by obstacles. Its meteorological and climatological measurements are continuous and date back to 1905. It is located in the stormiest region in the Czech territory with about 3.2 CG flashes/km$^2$/year (Novak and Kyznarova, 2020; Fig. 9a).

For registration of particles we use the Space Environment Viewing and Analysis Network (SEVAN) detector described in detail by Chilingarian et al. (2009). The basic SEVAN unit is composed of standard slabs of 50 x 50 x 5 cm$^3$ plastic scintillators. Between two identical assemblies of 100 x 100 x 5 cm$^3$ scintillators (four standard slabs) are located two 100 x 100 x 5 cm$^3$ lead absorbers and in the middle there is a thick 50 x 50 x 25 cm$^3$ scintillator stack (five standard plastic scintillator slabs). Scintillator light capture cones and photomultiplier tubes are located on the top, bottom and in the intermediate layers of the

detector. The slabs are sealed in a box made of 1 mm thick steel plate. The events described in this study were detected by the middle plastic scintillator stack of SEVAN, which was installed without the shielding lead absorber inside the building of the Milešovka observatory. The majority of the incoming increased radiation came through a concrete wall and a nearby window (see the detailed simulation results in Section 7). The energy threshold for the photomultiplier was set between 6.5 and 7.5 MeV. The counts are stored with a 1-min cadence. The energy of individual particles is not measured.


The vertical electrostatic field is measured by the electric field mill EFM 100 manufactured by the Boltek Company. The field mill is installed in an inverted position to minimize the noise originating from precipitation. The electric field is sampled at a cadence of 50 ms. Negative values at the field mill output correspond to an upward pointing electric field in which the electrons are accelerated downward. Two perpendicular broadband magnetic loop antennas SLAVIA (Shielded Loop Antenna with a

Versatile Integrated Amplifier) are used to measure the time derivative of variations of the horizontal magnetic field from 5 kHz up to 90 MHz (Kolmasova et al., 2018, 2020, 2022). The gain of the integrated preamplifiers is remotely controlled. The SLAVIA sensors are coupled with a digital oscilloscope sampling at a frequency of 200 MHz and the digitized signal is numerically integrated. The broadband analyzer is working in a triggered mode based on a predefined amplitude threshold: when it receives a trigger, it records a 168-ms long waveform snapshot including a history of 52 ms before the trigger. The

trigger time is assigned by the GPS receiver with an accuracy of 1 μs. The analyzer triggers on strong signals emitted by different lightning phenomena as return strokes (RS), intracloud (IC) discharges, or preliminary breakdown pulses. In case of a close thunderstorm, it also triggers on very fast sub microsecond pulses radiated by corona-type discharges occurring at the

tips of close metallic objects due to the strong electric field below the thundercloud. In this study, we use the measurements of the antenna oriented in the east-west direction.


The vertically oriented cloud radar was installed at the Milešovka observatory in 2018. It is a Doppler polarimetric radar (MIRA 35c), which was manufactured and installed by METEK Gmbh (http://metek.de/). It transmits electromagnetic signal within the Ka-band with a center frequency of $35.12 \pm 0.1$ GHz and a peak power of 2.5 kW. The radar core is of a magnetron type and the radar antenna is of the Cassegrain type with a diameter of 1 m, a gain of 48.5 dB and a beam width of $0.6°$. The

pulse repetition frequency varies from 2.5-10 kHz and the pulse width from 100 to 400 ns. The unambiguous velocity range ($\pm$ VNyquist) is $\pm 10.65$ m/s. The radar registers Doppler spectra, which correspond to averages of 40 consecutive values above the noise floor. The values below the estimated noise floor are deemed to have no signal. The internal software of the radar provides three moments of the Doppler spectra, such as radar reflectivity (Z), Doppler vertical velocity (DVV) and spectrum width, and derives other quantities such as the linear depolarization ratio (LDR) or signal-to-noise ratio. The temporal

resolution of the cloud radar is approximately 2 s, while its vertical resolution covers 509 gates, which are distant by 28.8 m one from another. The relatively narrow melting layer can be often detected in the radar reflectivity plots as a region with enhanced reflectivity, due to sudden changes in the hydrometeor properties (shape, size, and melting fraction) at temperatures below and above $0°C$ (Ryzhkov and Zrnic, 2019). The method of the hydrometeor classification used in this study was described by Sokol et al. (2018) and its refined version was reported in Sokol et al. (2020). Prior to the hydrometeor

classification, we correct the DVV values using the de-aliasing procedure and estimate the vertical air velocity (VAV). The calculation of VAV is based on a common approach according to which the very small particles (i.e. tracers) are so light that they are considered to be carried by the air only, which means that their velocity determines the VAV (Kollias et al., 2001; Gossard, 1994; Shupe et al., 2004). The hydrometeor classification assumes that the terminal velocity varies from one hydrometeor class to another and the hydrometeor classes naturally depend on the ambient air temperature. The classification

scheme uses the information about the altitude of the melting layer. Below the melting layer, snow or ice cannot exist because they have small terminal velocities and almost immediately melt in the melting layer or just below it. Therefore, only graupel, hail, cloud droplet and rain can appear between the ground level and the melting layer. Thus, based on the ambient air temperature, on the terminal velocity range of hydrometeors and on the shape of particles determined by LDR, five hydrometeor classes are distinguished in our classification: cloud liquid water, rain, graupel, hail, and ice/snow (Sokol et al.,

2020). Based on this classification, we can suggest areas where cloud electrification occurred; however, our radar does not directly measure the charge structure of the cloud. It is not a fully polarimetric radar and does not measure quantities like KDP (Differential Reflectivity) or ZDR (Specific Differential Phase), which were used, together with the lightning mapping array data, for example by Biggerstaff et al. (2017) to retrieve the locations of charge centres.

## 3 Thunderstorms on 23 April 2018

A cold front belonging to a pressure low over the Norwegian Sea was travelling to Central Europe and replaced a warm central European air mass by a cool maritime polar air. During the day, the CAPE (Convective Available Potential Energy) values gradually increased from almost zero at 00UTC to roughly 800 J/kg at Prague-Libuš (CZ), Mainingen and Kümmersbruck (both DE) sounding stations. The CAPE increase was accompanied by CIN (Convective Inhibition) decrease, which supported the evolution of convective storms. According to the radar measurements from the CZRAD network operated by the Czech

Hydrometeorological Institute (Novák, 2007) and EUMETSAT satellite measurements, the morning storms in Germany produced a nicely evolved gust front, which produced a squall line crossing the northwest of the Czech Republic at midday. The origin of the squall line was supported by a direct hodograph with almost no directional shear and a considerable deep layer shear (0-6 km) around 15 m/s. The storms crossing the Milešovka observatory in the afternoon were also linearly organized, however, they did not evolve into a squall line like the morning storms. The thunderstorms described in this study

occurred around noon (from 10:40 to 13:20 UTC) and in the evening (from 17:00 to 17:50 UTC) and we respectively marked them "storm A" and "storm B". The linear organization of both storms is clearly visible in Fig. 1, where black crosses in both panels show the location of the Milešovka observatory.

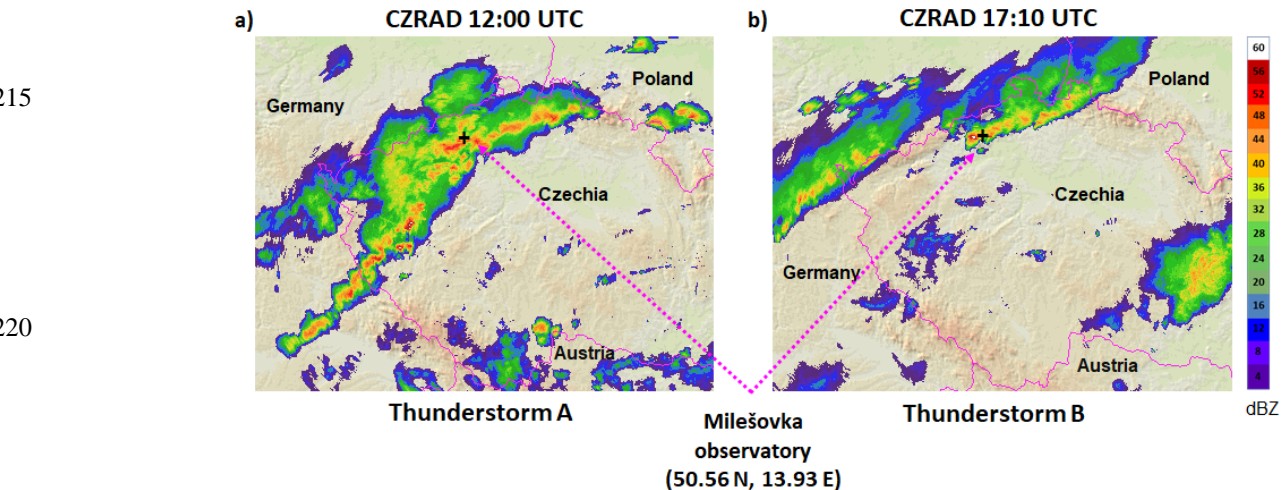

**Figure 1**: Maps of radar reflectivity for the most intense parts of storm A (a) and storm B (b). Black crosses show location of the Milešovka observatory (source www.chmi.cz on 23 April 2018).

The information about the temperature, relative humidity, air pressure, wind speed and its direction, precipitation totals, duration of sunshine and other meteorological parameters is available from the measurements of the automatic Vaisala weather

station. For our study, we use the precipitation totals measured in a 1-min cadence, which are shown by blue lines in Figs. 2a and 3a. To estimate an altitude of the cloud base we assume that it in simplicity corresponds to the lifted condensation level

(LCL) (Daidzic, 2019), which we calculated from the temperature at a level of 2 meters and the dew point temperature. Note that this estimation is quite rough as during the precipitation the calculation might be influenced by an increased relative humidity and decreased temperature. The LCL height represents the altitude of the lowest possible cloud base and the error in the LCL height estimation when using this simple method could reach 15 % (Lawrence, 2005). Red stars in Figs. 2a and 3a display the altitudes of the LCL above the Milešovka Mt. during both storms. (Note that all altitudes in Figs. 2 and 3 are relative to the altitude of the Milešovka station of 837 m a.s.l). The altitude of the cloud base was estimated to decrease from 1100 to 200 m above the station during storm A. During storm B, the height of the cloud base varied between 180 and 240 m. The 0° C level was located at an altitude of about 2 km above the cloud radar.

## 4 Particle measurements

The middle scintillator of the SEVAN detector, which was placed close to the window inside the observatory building, detected around 3500 counts/minute during the undisturbed conditions.

The count enhancements observed during storms A and B are shown in Figs. 2a and 3a, respectively by black lines. During storm A, the particle counts started to grow at 11:05 UTC, reached a maximum of about 4600 counts/min in 10 minutes, and dropped in the next 10 minutes nearly to the normal count rate. This significant increase of 31 % was later followed by two weaker enhancements of 18 % and 14 %. The fluctuations of the count rate lasted for about 70 minutes. During storm B, the count rate started to grow at 17:10 UTC, reached a maximum of about 5200 particles per minute at 17:23 UTC and dropped to the normal count rate at 17:33 UTC. The maximum count increase was unusually large, reaching 48 %. The precipitation rate shown by blue lines in Figs. 2a and 3a started about 10 minutes after the count increase during storm A (Fig. 2a) and about 3-4 minutes later than the count increase in case of the storm B (Fig. 3a). The estimated cloud base was respectively found 200-1100 m and 180-240 m above the observatory, during the count increases in storm A (Fig. 2a) and storm B (Fig. 3a). The most intense parts of the TGE events happened when the cloud base was located at about 800 m during storm A and at about 200 m during storm B.

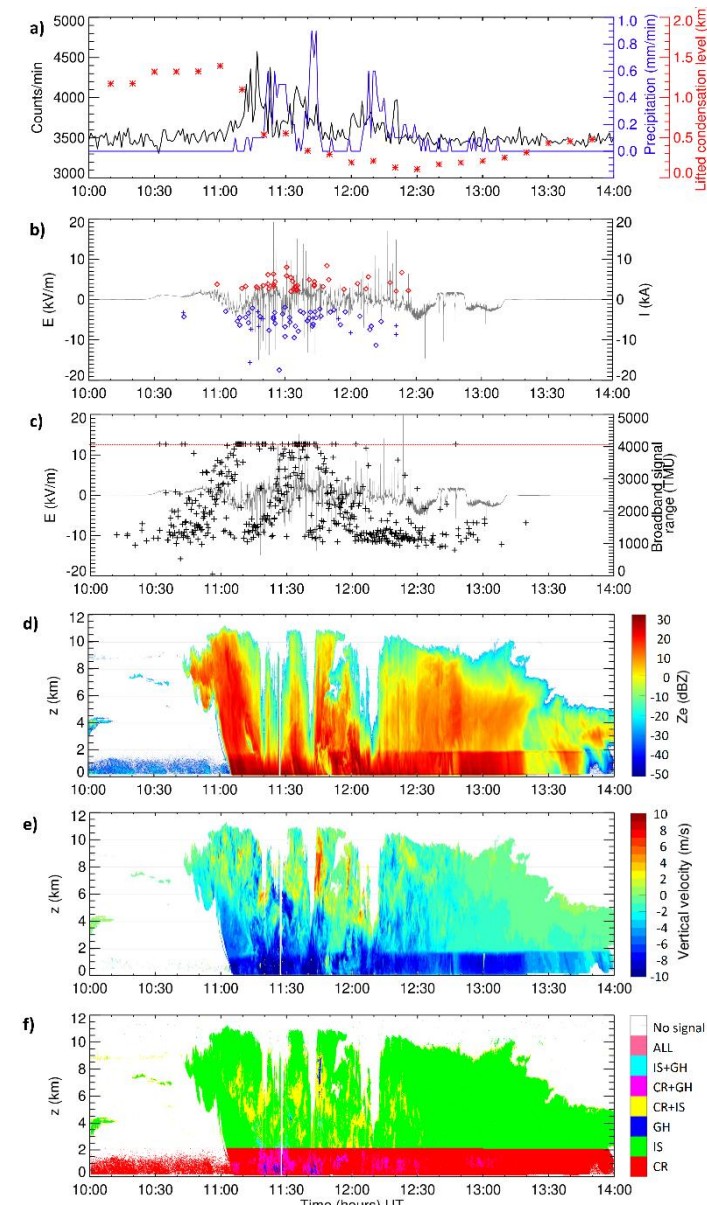

**Figure 2**: Storm A: a) Particle counts per minute (black line), precipitation totals in mm/min (blue line), the altitudes of the lifted condensation level in km above the altitude of 837 m - Milešovka observatory (red stars); b) fluctuations of the vertical electric field measured by electric field mill (grey line), EUCLID detections: red and blue color for positive and negative discharges, diamonds for IC discharges and crosses for CG discharges; c) fluctuations of the vertical electric field (grey line), absolute maximum of the range of values measured by the broadband antenna (in telemetry units, each black cross corresponds to the maximum range of the magnetic field derivative recorded during one 168 ms long waveform snapshot), red line shows

the saturation in both positive and negative polarity (4096 TMU); d) the radar reflectivity; e) vertical updraft velocity; f) classification of hydrometeors (G-graupel, H-hail, I-ice, S-snow, C-cloud water, R-rain).

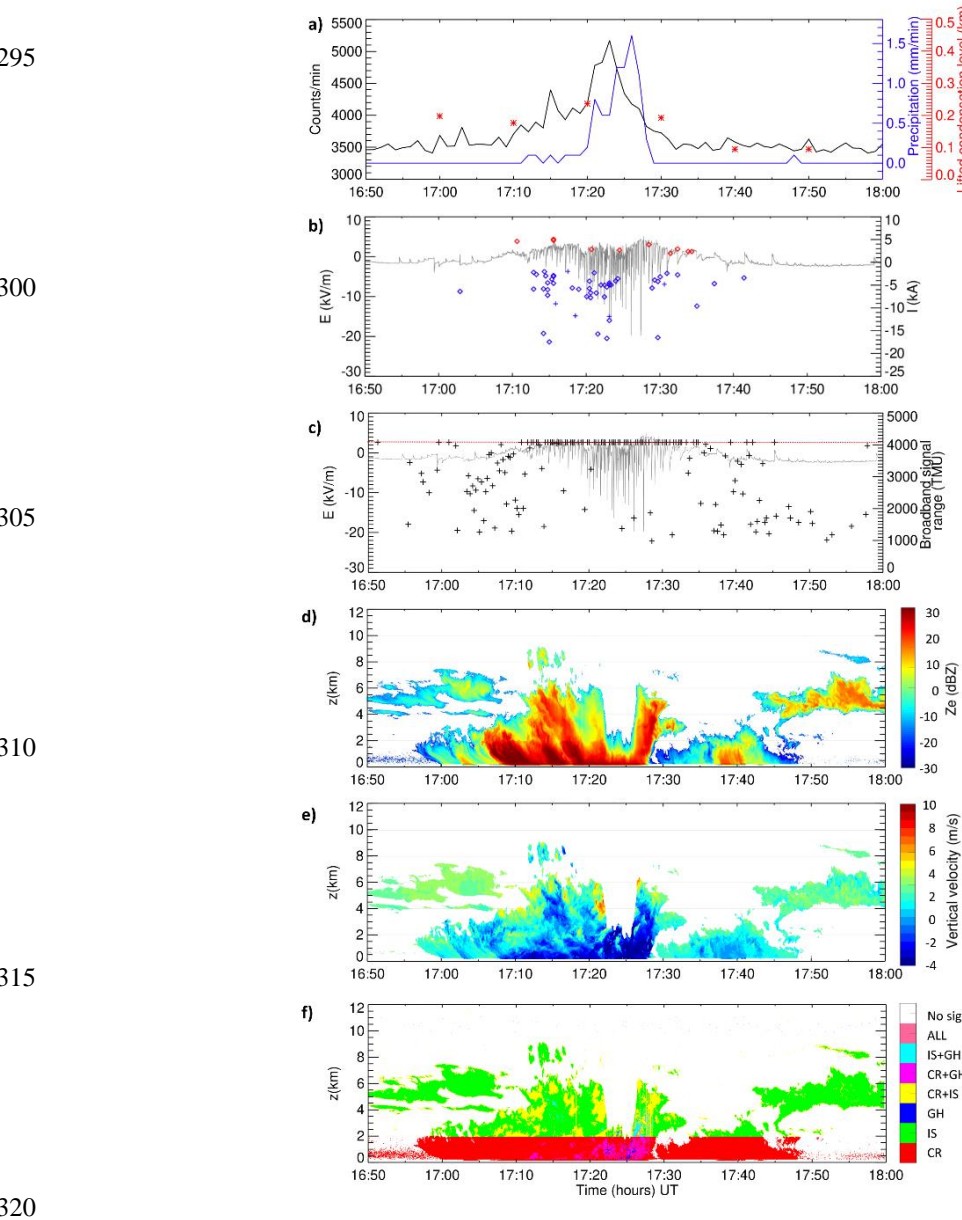

**Figure 3**: The same as in Fig. 2 but for the storm B.

## 5 Electromagnetic measurements

### 5.1 Electric field mill measurement and EUCLID detections

Variations of the atmospheric electric field measured by the electric field mill during the investigated events are shown by grey lines in Figs. 2b, c and 3b, c, respectively. The field mill data show small values of the electric field until 10:45 and since 13:10 during storm A due to the fair-weather current flowing from the ionosphere to the ground. In case of storm B, the electric field waveform exhibited small peaks also before 16:55 and after 17:45. These peaks can be assigned to lightning discharges that occurred more than 5 km away from the sensor. Significant variations of the electric field were detected when the thundercloud was located above the field mill from 10:45 to 13:10 UTC, and from 16:55 to 17:45 UTC, respectively, during storms A and B. The maximum values reached ± 20 kV. Surprisingly, negative peaks in the electric field records dominated until 11:25 during storm A, and only negative pulses were observed during the whole period of storm B. Negative electric field excursions were very short and followed in a quick succession. Such variations of electric fields near the ground are not typical. Rapid changes of polarity in the otherwise slowly varying atmospheric electric field usually correspond to the neutralization of the charge in the thundercloud due to close IC or CG discharges. Detections of the European lightning location network EUCLID, limited up to 5 km from the Milešovka observatory, are displayed in Figs. 2b and 3b by colored symbols. Red and blue crosses are used for positive and negative CG discharges, respectively. Red and blue diamonds show positive and negative IC discharges. The negative sign indicates the movement of the negative charge downward. A substantial lack of CG discharges may be noted. The negative IC discharges, called also inverted IC flashes, were clearly associated with periods of negative excursions of the atmospheric electric field and their occurrence can thus be considered as the primary case of the observed negative peaks. However, inverted IC flashes were detected by EUCLID only for some of observed negative peaks. The amplitudes of negative peaks also clearly do not correspond to the reported peak currents of the inverted IC flashes. Note that the IC/CG classification accuracy depends on the polarity and strength of the discharge and reaches about 80%, while the misclassified strokes were generally very weak (Schwalt et al., 2020).

### 5.2 Broadband magnetic field measurement

The trigger of the broadband analyzer was nearly constantly activated during the time when the thundercloud was located above the observatory. The cadence of the 168-ms long waveform snapshots was mainly given by the ability of the analyzer to store and transfer the data. The maximum amount of 3-4 snapshots per second was recorded during the time of the intense variations of the atmospheric electric field, when the limitations of the throughput of our acquisition system were reached. 474 snapshots sampled at 200 MHz were recorded from 10:12 to 13:20 during storm A. During storm B, a set of 159 snapshots was recorded from 17:35 to 17:58.

An example of a waveform snapshot is shown in Fig. 4. The panel 4a displays the complete numerically integrated waveform; panel 4b shows the raw signal measured by the antenna. The waveforms include also a history of 52 ms recorded before the

trigger. After the time of the trigger, the signal frequently reached the digital saturation of ± 2040 telemetry units (TMU), when one TMU corresponds to 76 µT/s. As the antenna measures the time derivative of the magnetic field, the repeated saturation indicates very fast changes of the magnetic field. A thorough look at the waveform details reveals that the records are composed from a mixture of pulses of different shapes, widths, and polarities. This mixture is not surprising: the analyzer is able to detect the RS pulses and IC pulses occurring a few hundreds of kilometers from their causative discharge and in case of a very close storm it is often saturated by fast pulses emitted by corona discharges appearing at the tips of close metallic objects. Fast unipolar pulses with a width of a few tens of nanoseconds which originated in close corona discharges are shown in Figs. 4c and 4f.

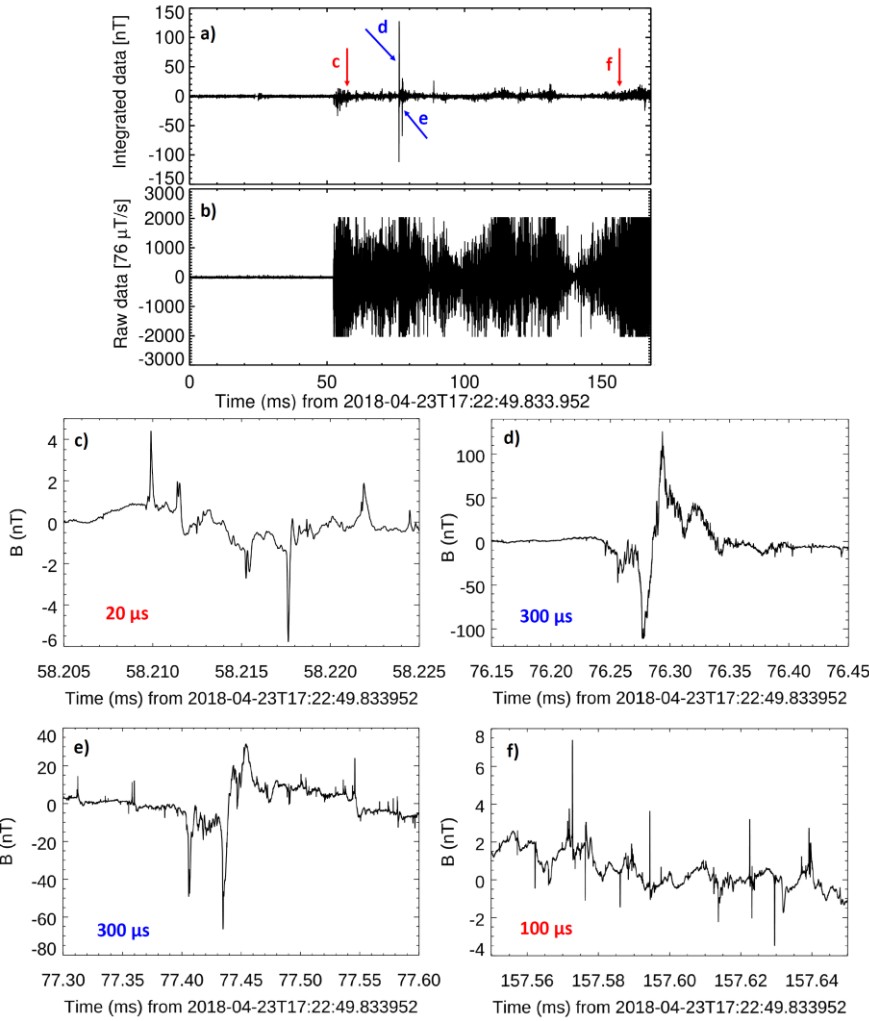

**Figure. 4:** Example of a 168-ms long magnetic field waveform snapshot measured by a broadband antenna: a) integrated magnetic field waveform; b) the time derivative of the magnetic field fluctuations, the trigger was activated at 52 ms; c-f) 20-

300 μs long details indicated by arrows in the panel a: c) submicrosecond unipolar pulses emitted by corona discharges; d) a bipolar pulse emitted probably by a strong close IC discharge; e) a RS type pulse; f) mixture of different pulses. Note that the numerically integrated values might be inaccurate because of a frequent saturation of the received signal.

The rapidly changing polarity either indicates that the corona discharges arose at different directions with respect to the axis of the magnetic loop antenna, or that both negative and positive corona discharges occurred. We are not able to distinguish between these two possibilities by our measurements. Fig. 4d shows an intense bipolar pulse at 76.24 ms emitted probably by a close IC discharge. A typical RS pulse is recognizable in Fig. 4e at 77.43 ms and might have been generated by a distant CG

discharge.

All waveform snapshots are composed of a mixture of pulses and a large part of snapshots recorded during active parts of both storms reached the negative, positive or both digital saturation levels. We therefore calculated the maximum range of measured raw values for all 633 waveform snapshots. The obtained values are plotted by black crosses in Figs. 2c and 3c. The maximum

range of 4080 TMU is represented by red solid lines. It is clearly visible that during intense negative excursions of the atmospheric field during storm B saturation was reached during nearly all recorded waveform snapshots.

## 6 Cloud profiler measurements

### 6.1 Storm A

Figure 2d shows the time evolution of the radar reflectivity factor, which we measured by the cloud radar, during the first

thunderstorm event (11-14 UTC approximately). It clearly depicts that the values of reflectivity, especially during the first half of the event, reaches or exceeds 30 dB in most of the vertical profile. The vertical extent is up to 10 km suggesting a vertically developed thundercloud situated above the Milešovka observatory. Based on high reflectivity values, we can assume that the melting layer was at a height of 2 km above the radar approximately and that up to 13:20 UTC precipitation occurred in lower elevations. This altitude of the melting layer corresponded to the value calculated from the measured ground temperature

(using a gradient of -6.5°C/km) and served as an input for the hydrometeor classification. The lower reflectivity values during the second half of the event represents the rear part of the squall line where slighter precipitation than in the front part is located, Figure 2e displays the time evolution of DVV, where the upward motion is depicted by positive values while the downward motion by negative values. Positive values, i.e. upward motion, prevails in upper elevations, while negative values, i.e. downward motion, dominates lower elevations and suggest fall out of greater (i.e. precipitation) particles. Hydrometeor

distribution as it resulted from the algorithm by Sokol et al. (2020) during the first thunderstorm event is depicted in Figure 2f. Naturally, most of the higher elevations consist of ice and snow, while the lower elevations consist of liquid cloud water and rain. However, it is worth noting that during the first half of the event, we also detected graupel and hail in the lowest elevations, which can be related to heavy precipitation. Interestingly, from about 5 to 9 km above the radar at 11:45 approximately, there

is a mixture of graupel and hail surrounded by ice and snow and supercooled liquid cloud water. This might be a place where the process of cloud electrification could be expected according to the widely accepted theory of cloud electrification by collisions of graupel with ice and snow particles in presence of supercooled cloud liquid water.

## 6.2 Storm B

Figure 3d shows the time evolution of the measured radar reflectivity by the cloud radar during the second thunderstorm event. As compared to the first thunderstorm event, it is obvious that the vertical extent of the cloud during the second thunderstorm event is much lower, up to 6 km above the radar approximately. This may be related to the fact that the cloud radar is only vertically pointing and thus does not see the whole thundercloud horizontally. In the case of this thunderstorm, the cloud radar likely scanned a side part of the thunderstorm instead of its core, as can be seen on Figure 1b. However, the radar reflectivity values are as high as during the first thunderstorm event, suggesting a possible fallout of precipitation. The melting height is hardly definable from the radar measurements in this case, so for the hydrometeor classification we have to calculate it from the measured ground temperature. The gap in measurements from 17:20 to 17:30 UTC from about 2 to 6 km can either be related to attenuation of the radar signal by heavy rain but the reflectivity values are pretty low or simply and more likely it corresponds to the fact that the cloud was not that extensive at that time. The time evolution of DVV is displayed in Fig. 3e. In contrary to the first thunderstorm event, the upward motion reaches lower values in general, while the downward motion is similar to the first event, suggesting precipitation fallout. As far as the hydrometeor distribution during the second thunderstorm event is concerned, Fig. 3f shows the time from 17:20 to 17:30 the most interesting as all the hydrometeor classes were detected at that time, though the cloud is too low to draw any conclusions out of it. Further, the distribution is expectable and similar to the first event with e.g., a predominance of cloud water and rain at the lowest elevations.

## 7 Simulation of particle fluxes

The installation of the particle detector SEVAN was not the same as for other measurements sites and the obtained counts thus are not directly comparable with the existing TGE reports. We therefore verify the enhancement of counts in our detector for a known TGE energy spectrum (Chilingarian et al., 2012) using the PHITS (Particle and Heavy Ion Transport code System) Monte Carlo based program for particle transport simulations. We use the version 3.24 released in 2021 together with the INCL, EGS5, and KUROTAMA models. We also used the PHITS Cosmic-ray source mode based on PARMA/EXPACS (Sato et al., 2018; Boudard et al., 2013; Iida et al., 2007; Sato, 2015, 2016).

The geometry of the detector installation inside the observatory building is simplified as an inner space of air with dimensions of 4 x 4 x 3 m in x, y, and z axes (where z is vertical pointing up), surrounded by concrete walls with a width of 80 cm. At a height of 1 m there is a 1 m high window opening with a width of 0.5 m. The opening is filled with a 2 mm thick glass. The sensitive volume of plastic scintillator with the dimensions of 50 x 50 x 25 cm3 is placed at a distance of 40 cm from the

window and 100 cm above the floor. The scintillator is covered with a 1 mm thick steel plate, which represents the scintillator box. On the top of the concrete ceiling, a wooden plate with a thickness of 3 cm and a steel plate with a thickness of 1 mm represent the roof. This environment is visible as a black rectangular shape in Fig. 5a.

As the first step, we test our setup by comparing measured and calculated background count rates originating in the secondary cosmic ray particles. The measured background rate was around 3500 counts/minute (Figs. 2a and 3a). The PHITS's cosmic

rays source for the specific date, height, and geometry gives us a total number of $3640 \pm 135$ counts/minute for the energy higher than 6.5 MeV deposited in the scintillator ($3432 \pm 118$ muons/minute, the rest include electrons/positrons, photons and neutrons). The calculated count fits well the observation.

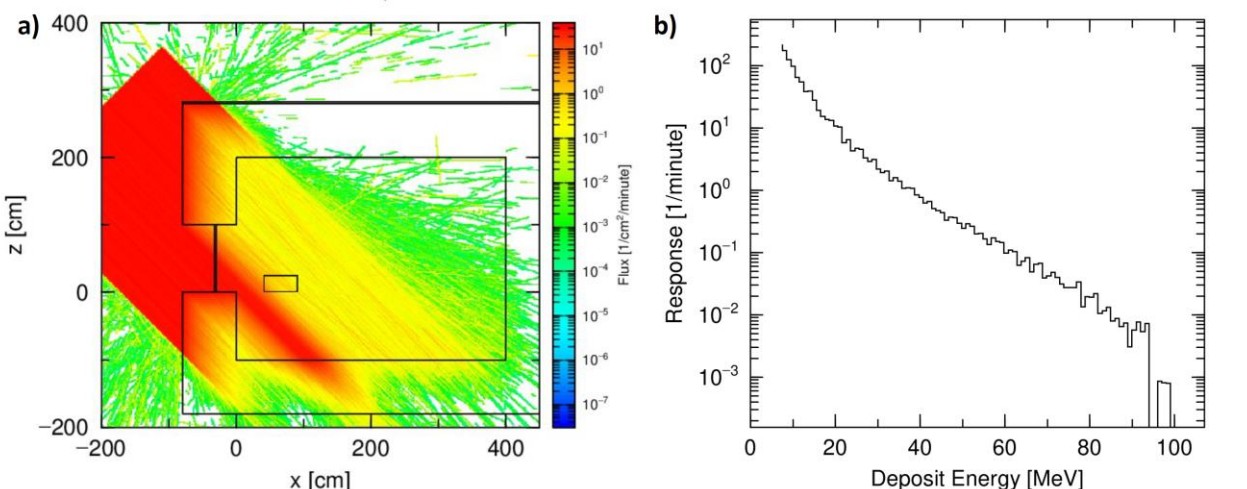

**Figure 5:** a) TGE source particles, the source is tilted by 45° from the vertical axis, and the energy spectra correspond to observation by Chilingarian et al. (2012), their Fig. 12. b) Calculated deposit energy spectra inside the scintillator. Lower

energy threshold was set to 6.5 MeV according to the setup of the detector. The detector itself does not provide the energy spectra.

As we do not measure the energetic spectrum for the TGE events observed at Milešovka, we use a known TGE spectrum measured at the Aragats observatory on 4 October 2010 (Chilingarian et al, 2012) when the count enhancement in the middle

SEVAN scintillator reached about 1400 counts/min, which is similar to our observation. The background level was about 7100 counts/min (http://adei.crd.yerphi.am/). The source geometry is represented by a square of 3 x 3 $m^2$ located at a distance of 3 meters from the detector in order to minimize the influence of scattered particles. The direction of the particle beam is perpendicular to the source plate and the detector is approximately in the middle of the beam. The dead time of the detector is not taken into account due to the fast response of the plastic scintillator and due to the very high energy threshold. The source

is composed only from photons. The electrons are not included, as their ability to cross the concrete walls or the window, and to deposit enough energy in the scintillator is negligible. According to Chilingarian et al. (2012), the photon differential

intensity $I$ in particles/min/MeV/m$^2$ can be represented as $I = 4.5 \times 10^5 \, e^{-(0.25 \, E/MeV)}$ for photon energies $E$ from 5 to 10 MeV, and $I = 6.3 \times 10^7 \, e^{-(3.3 \, E/MeV)}$ for photon energies $E$ from 10 to 100 MeV. By integrating the spectrum over all energies, we get the total number of photons in the source equal to 5.05 x 10$^5$ per minute and m$^2$. If we place the photon source

on the top of the simulated environment, slant it by 45° to let the particles enter the window, the area of the detector (small box at coordinates x=50, z=0 in Fig. 5a) is hit by approximately one source particle/cm$^2$/minute. The surface of the detector perpendicular to the beam is 2652 cm$^2$. The spectrum of particle energies absorbed in the scintillator and originating solely from the TGE source (without cosmic rays) is shown in Fig.5b. The count rate of particles with an energy range from 6.5 to 100 MeV deposited in the detector can be estimated using the T-deposit tally as $(977 \pm 3)$/min. This number represents a 27%

increase of the count rates relative to the background, which is roughly consistent with the observed peak count rates (31% and 48%, respectively, for storm A and storm B in Fig. 2a and 3a). We have calculated the count enhancements also for other inclinations of the beam. We verified that the enhancements calculated for an inclination of 45° best reproduced our measurements. To obtain the same values for a beam arriving more vertically to the detector we would need to assume a stronger TGE. A more horizontal inclination of the beam seems to be not realistic.

**7 Discussion and summary**

Two significant TGE events were registered on 23 April 2018 at the Milešovka observatory in Czechia at 837 m a.s.l. To our best knowledge, it was the first multi-instrument TGE observation below 1 km above the sea level outside Japan. The registered increases of photon count reached 31 % and 48 % above the background level during the thunderstorms A and B, respectively. The duration of TGEs was unusually long in comparison with other reports (Chum et al., 2020; Chilingarian et al. 2017 a,b,

2020b), lasting about 70 minutes (storm A) and about 25 min (storm B). The increased counts were detected by a plastic scintillator of the particle detector SEVAN (Chilingarian et al., 2009). Rain appeared with a delay of several minutes after the count increases during both events.

This delayed rain arrival and an energy threshold of 6.5 MeV for registered particles in the scintillator clearly exclude the

presence of the radon progeny washout and its subsequent decay in the count rates. Using the simulations, we have shown that the observed increases of count rates might have been related to TGEs. We also verified that no extreme cosmic ray events were detected during these observations (https://gle.oulu.fi/#/).

A question, however, remains, why only these two TGE events were registered at the Milešovka observatory, while the particle

measurement was operational also in the thunderstorm seasons 2020 and 2021. Based on long-term observations (Kašpar et al., 2018), meteorological data from 23 April 2018 do not indicate any extreme weather event. The TGE events were observed during two convective storms with well-organized multi-cells, but neither precipitation totals (Fig. 2a, 3a) nor the maximum wind speed of 14 m/s exceeded values observed during numerous thunderstorms occurring in the same area in 2020 and 2021.

The observatory was not inside the thundercloud as it was the case of TGEs observed at the high mountain observatory (Chum et al., 2020). The cloud base was respectively found at least 200 m and 180m above the observatory during the storm A and B. The 0°C level was located at an altitude of about 2 km above the cloud radar. An estimated height of the cloud tops during the storm A was about 11 km (Figs. 2d-f), as expected in the midlatitudes. The cloud tops during storm B were lower, at about 8 km (Figs. 3d-f), indicating that the storm center of the second storm was not directly above the cloud radar. The updraft velocities reached typical values of 10 m/s. During both storms, the cloud radar recognized graupel below the melting level. Based on the classification of the hydrometeors using the Ka-band cloud radar data, the composition of hydrometeors suggested good but not extreme conditions for the cloud electrification.

Analysis of electromagnetic measurements (broadband magnetic-field data, records of the vertical electric field monitor and EUCLID lightning detections) reveals several interesting characteristics of the investigated thunderstorms. Variations of near surface electric fields observed during both storms were very different compared to previous observations during TGE events, reported for example by Chum et al., (2020), Chilingarian (2017b, 2020). In our case, the data showed a completely untypical behaviour: negative electric field excursions were very short and followed each other in a quick succession. Some of them were associated with inverted IC strokes detected by EUCLID, none of which, however, abruptly reduced or terminated the TGE flux. We cannot exclude that short duration TGE events of a few tens of seconds could have been reduced (Kochkin et al, 2021) or terminated (Chum et al., 2020; Chilingarian, 2017b, 2020) by a lightning stroke, as these would not be recognisable in the 1-min cadence SEVAN data. A frequent occurrence of IC lightning with and a low occurrence of CG lightning indicate a presence of a strong lower positive charge region. The increases of the TGE radiation corresponded in time to the frequent occurrence of large negative pulses electric fields (up to – 20 kV/m) corresponding most probably to inverted IC strokes. We also identified numerous sub-microsecond-scale pulses in the broadband magnetic field records, which often saturated the preamplifier (Fig. 4b), and can be attributed to corona-type discharges occurring at close metallic objects near the receiving antenna in high local electric fields below the thundercloud. Note that visible sparks were not expected to be reported by the observatory staff during the daytime. These fast unipolar, a few tens of nanoseconds wide pulses (shown in detail in Figs. 4e) are similar to pulses emitted by corona discharges in Arcanjo et al. (2021). Unlike Arcanjo et al. (2021) we cannot distinguish between pulses emitted by positive and negative corona, as their polarity is dependent not only on the direction of the corona current but also on the relative orientation of the magnetic loop to the source discharge. Based on the simulation by Kašpar et al. (2015) the unipolar character of pulses indicates a high propagation velocity of the current waves or short discharge channels or both, which is consistent with expected properties of corona-type discharges.

All electromagnetic measurements indicate a presence of a strong LPCR:

a.      an increased occurrence of inverted IC lightning,

b.      a suppressed occurrence of CG lighting,

c.      a presence of irregularly distributed narrow unipolar pulses linked to strong corona discharges which might have been contributing to the delivery of additional positive charge to the cloud base.

Moreover, the cloud radar identified graupel, which are supposed to carry positive charge at temperatures above the melting
level (Takahashi, 1978). LPCR inside the thundercloud is probably responsible for an adequately high electric field in the lower thundercloud dipole between LPCR and the main negative charge region extending over at least 2 km, as we can estimate from the hydrometeor classes observed by the cloud radar. This extended charge structure was probably capable to accelerate seed electrons and as a result, we observed significant long-lasting bremsstrahlung.

The exceptionality of the observation rises a question if the secondary cosmic ray particles, which might have been
substantially attenuated at the altitude of the observatory, were the only source of the seed electrons. It is possible that substantial a part of the seed electrons might have originated in a high concentration of radon in the air collected above the uranium-rich soils during rainless period before the thunderstorms overpassed the area (https://remap.jrc.ec.europa.eu/Atlas.aspx#).

In summary, our multi-instrument data recorded during two continental thunderstorms on 23 April 2018 reveal that TGEs can occur at an altitude lower than 1 km above the sea level. The uniqueness of these TGE registrations implicates the idea that the observational conditions might have been unusually favourable. The meteorological situation allowed for a formation of a strong lower positive charge region with its lower edge located close to the observatory assuming the lower edge of the LPCR was located at the cloud base at the beginning of the storm (Rakov and Uman, 2003).

The altitude of the cloud base varied between 1100 and 200 m above the observatory during the storm A and between 240 and 180 m during the storm B. Nevertheless, the LPCR is a transient phenomenon, which is moving down with positively charged falling graupels. Therefore, it is probable, that the LPCR might have been located even closer to the detector during the graupel fall, when we observed the particle flux maxima. The presence of lower positive charge region was indirectly confirmed by the electromagnetic measurements, with a possible contribution from local corona discharges, and by the cloud radar data. The

lower thundercloud dipole was probably capable to accelerate the seed electrons in the air. These seed electrons might, besides the usually considered source from the secondary cosmic rays, also originate from radon in the air collected in this specific region. A follow-up study is needed to test the absence of a large LPCR in other storms without recorded TGEs.

**Data availability**

The electromagnetic data are available at http://bleska.ufa.cas.cz/ml/. Particle data are available at http://sevan.ufa.cas.cz/.

**Author contribution**

IK and OS designed the study and interpreted the results. JS and OP analysed the particle data and performed the flux simulations. JP and ZS analysed and interpreted the cloud profiler data. PZ described the meteorological context. GD provided

the EUCLID data. RaL, RoL and IS were responsible for the instrumentation and data storage. The paper was written by IK, OS, JP and PZ.

## Competing interests

The authors declare that they have no competing interests.

## Acknowledgements

The work was supported by the GACR grant 20-09671S and by the European Regional Development Fund-Project CRREAT (CZ.02.1.01/0.0/0.0/15_003/0000481).

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
