# Peer review of "Continental Thunderstorm Ground Enhancement observed at an exceptionally low altitude"

_Atmospheric Chemistry and Physics, 2022_

## Referee Comment (RC1)

*Referee comments on the paper: " Continental Thunderstorm Ground Enhancement observed at an exceptionally low altitude" by Ivana Kolmašová, Ondřej Santolík, Jakub Šlegl et al.*

**The paper presents one of the first observations of the TGE at considerably low altitude with the same particle detector used in earlier experiments for the detection of numerous TGEs at high-mountain observatories on Aragats, Musala, and Lomnicky Stit. It is another confirmation that TGE is a universal physical phenomenon, that established high-energy physics in the atmosphere as a new synergetic scientific direction between Cosmic Ray and Atmospheric physics. The presented evidence is very important as particle measurements were accompanied by the measurements of atmospheric parameters, such as near-surface electric field disturbances, lightning detection, hydrometeor location with radars, and broadband magnetic field measurements. Analysis of these multivariate data demonstrates the emergence of low located LPCR, which leads to electron acceleration by the strong electric field between the main negatively charged layer in the middle of the thundercloud and low located transient LPCR.**

**I recommend the paper for publication after answering my questions below.**

**Most important is to demonstrate with better Figures and better arguments that the LPCR height is about 200 m, as was declared in the paper.**

"The main complications for observations of TGEs were (a) emissions originating in the decay chain of the radon washed out from the air by rain. "

**Sure, radiation of 226Rn progeny (mostly 214Br and 214Pb) can be overwhelming for the low energy (up to 1.5-2 MeV) energies. However, this sentence does not explain how the Radon and its progeny occurred in the air. Experiments on Aragats show that Radon and its progeny are after attaching to charged aerosols are lifted by the near-surface electric field (NSEF) to the air and their radiation is registered by particle detectors simultaneously and long after the TGE particles. If it is rainy, the rain will return some of the isotopes back to the ground, if not, as usually during TGEs, the radiation from the air will be continued for 1-2 hours till Radon progeny finally decays [1].**

"TGEs were rarely detected also during positive electric fields. This can be explained by the asymmetry in the electron/positron component of the secondary cosmic rays (Zhou et al., 2016)."

**Please, add some explanations for how this asymmetry influences the polarity of the electric field. If NSEF is positive for several minutes, it means a large positive charge above (LPCR) that "screened" a larger main-negative charge in the middle of the cloud (MN), see scenarios of TGE origination in [2], Fig.1**

"…but sometimes they are terminated abruptly by a nearby lightning discharge",

**I recommend stronger expression, not sometimes, but quite often, see [3].**

" Electromagnetic pulses emitted by corona discharges might be identified in fast electromagnetic recordings. "

**Can you, please, make additional clarifications? Your argument is "microsecond duration unipolar pulses emitted by corona discharges" only or something additionally?**

"none of which, however, abruptly terminated the TGE flux."

**At very intense lightning activity TGEs are lasing a few tens of seconds before being interrupted by the lightning flash. The 1-minute time series smooths the count rate surges, which are seen only on 1-s time series of count rate.**

"can be attributed to corona-type discharges occurring at close metallic objects near the receiving antenna in high local electric fields below the thundercloud."

**Can you localize where these discharges occurred? Do you have any report from staff seeing the corona discharges? Is it possible?**

"c. a presence of strong corona discharges which might have been contributing to the delivery of additional positive charge to the cloud base. "

**This conclusion comes from the "the unipolar character of pulses "only?**

"LPCR inside the thundercloud is probably responsible for a high electric field in the bottom thundercloud dipole, which accelerated seed electrons and as a result, we observed significant long-lasting bremsstrahlung."

Why you emphasize the bottom of dipole and not the whole dipole between Main Negative layer and LPCR?

**Have you estimated the MN height from the radar data? The extension of strong electron accelerating field should be 1-2 km for TGE initiation. Please, give an estimate of the extent of the" high electric field" you mentioned. It will be good if you estimate roughly the size of the lower dipole between MN and LPCR.**

"The meteorological situation allowed for a formation of a strong lower positive charge region located close to the observatory (only 180 m above it during the storm B). "

**Please, explain in more detail how you obtain the estimate of 180 m., please, give the height in limits, usually, LPCR extension is several hundred meters and it is a transient phenomenon, changing and finally escaping with graupel fall.**

**References**

1. Chilingarian, A., Hovsepyan, G., & Sargsyan, B. (2020). Circulation of Radon progeny in the terrestrial atmosphere during thunderstorms. *Geophysical Research Letters*, *47*, e2020GL091155. https://doi. org/10.1029/2020GL091155.
2. A. Chilingarian, G. Hovsepyan, E. Svechnikova, and M. Zazyan, Electrical structure of the thundercloud and operation of the electron accelerator inside it, Astroparticle Physics 132 (2021) 102615 https://doi.org/10.1016/j.astropartphys.2021.102615
3. Soghomonyan, Suren; Chilingarian, Ashot ; Khanikyants, Yeghia (2021), "Dataset for Thunderstorm Ground Enhancements terminated by lightning discharges", Mendeley Data, V1, doi: 10.17632/p25bb7jrfp.1

---

## Referee Comment (RC3)

[referee-annotated manuscript omitted]

---

## Author Comment (AC1)

**We thank Ashot Chilingarian for careful reading of the manuscript and for his helpful comments and suggestions. We responded to all of them and revised the manuscript accordingly.** (The line numbers are related to the manuscript with tracked changes.)

**Responses to Reviewer (in blue) with Reviewer´s comments in black.**

Most important is to demonstrate with better Figures and better arguments that the LPCR height is about 200 m, as was declared in the paper.

As direct measurements of electrification in the thundercloud are extremely rare they are of course not available for our case. We therefore have to rely on realistic assumptions based on previous measurements, namely that the bottom of the thundercloud determines the position of the lower edge of LPCR (Rakov & Uman, 2003) and that the bottom of the thundercloud corresponds to the lifted condensation level (Daidzic, 2019). We also have other indirect indications of a strong LPCR at a low altitude above the observatory: the presence of strong corona discharges (Nag & Rakov, 2009; Chauzy & Soula, 1999), an increased occurrence of inverted IC lightning, a suppressed occurrence of CG lighting (Nag & Rakov, 2009), as stated on lines 527-531. To bring better arguments and to express better the limitation of our measurements we reworded the sentence in the summary on lines 546-548 which now reads as follows:

 *"The meteorological situation allowed for a formation of a strong lower positive charge region with its lower edge located close to the observatory (only 180 m above it during the storm B assuming the lower edge of the LPCR is located at the cloud base (Rakov and Uman, 2003))."*

*"The main complications for observations of TGEs were (a) emissions originating in the decay chain of the radon washed out from the air by rain. "*

Sure, radiation of 226Rn progeny (mostly 214Br and 214Pb) can be overwhelming for the low energy (up to 1.5-2 MeV) energies. However, this sentence does not explain how the Radon and its progeny occurred in the air. Experiments on Aragats show that Radon and its progeny are after attaching to charged aerosols are lifted by the near-surface electric field (NSEF) to the air and their radiation is registered by particle detectors simultaneously and long after the TGE particles. If it is rainy, the rain will return some of the isotopes back to the ground, if not, as usually during TGEs, the radiation from the air will be continued for 1-2 hours till Radon progeny finally decays [1].

As suggested, we added an explanation how the radon and its decay chain appear in the air and added the reference [1]. The text now reads as follows on lines 55-58:

„*The origin of radon and its progeny in the air was explained (Chilingarian et al., 2020a) by their attaching to charged aerosols after being lifted by the near surface electric field to the air. Their radiation then can be registered by particle detectors simultaneously with the TGE particles. Rain quickly returns some of the isotopes back to the ground. In the absence of rain, the radiation from the air can continue for 1-2 hours till radon progeny finally decays."*

*"TGEs were rarely detected also during positive electric fields. This can be explained by the asymmetry in the electron/positron component of the secondary cosmic rays (Zhou et al., 2016)."*

Please, add some explanations for how this asymmetry influences the polarity of the electric field. If NSEF is positive for several minutes, it means a large positive charge above (LPCR) that "screened" a larger main-negative charge in the middle of the cloud (MN), see scenarios of TGE origination in [2], Fig.1.

The hypothesis on the asymmetry in the electron/positron component of the secondary cosmic rays was introduced by Zhou et al. (2016) and later developed by Bartoli et al. (2018) using their TGE

observation at an altitude of 4300 m a.s.l. in Tibet, with a support of their Monte Carlo modelling. We did not observe any TGE during the positive near surface electric field periods, so we limit ourselves to briefly mentioning such observations (Kudela et al., 2016; Bartoli et al, 2018, Chum et al., 2020) which are unrelated to our case. The sentence on line 84 ow reads as follows:

*"TGEs were occasionally detected also during positive electric fields (Zhou et al. 2016; Kudela et al., 2016; Bartoli et al., 2018, Chum et al., 2020)."*

*"…but sometimes they are terminated abruptly by a nearby lightning discharge",* I recommend stronger expression, not sometimes, but quite often, see [3].
We changed the expression to "quite often" on line 86 and added the suggested reference [3].

*"Electromagnetic pulses emitted by corona discharges might be identified in fast electromagnetic recordings. "*
Can you, please, make additional clarifications? Your argument is "microsecond duration unipolar pulses emitted by corona discharges" only or something additionally?
We added a more detailed description of the properties of pulses which might be generated by corona discharges. The wording on line 111-116 is now as follows:

*"Electromagnetic pulses emitted by corona discharges might be identified in fast electromagnetic recordings from their microsecond duration, unipolarity, and random distribution (Arcanjo et al., 2021). Unipolar microsecond scale pulses themselves were also found to accompany in-cloud processes as dart leaders or K-changes, but these appeared in several hundred microsecond long pulse trains with regular inter-pulse intervals (Rakov et al., 1992; Kolmašová and Santolík, 2013). Therefore, these pulse can be distinguished from the characteristic radiation from local corona discharges observed in electromagnetic recordings."*

*"none of which, however, abruptly terminated the TGE flux."*
At very intense lightning activity TGEs are lasing a few tens of seconds before being interrupted by the lightning flash. The 1-minute time series smooths the count rate surges, which are seen only on 1-s time series of count rate.
We added (lines 513-514) a sentence stating the limitations of our SEVAN setup to recognize short TGEs.

*"We cannot exclude that there were short duration (a few tens of seconds) TGE events terminated by a lightning stroke, as these would not be recognizable in the 1-min cadence SEVAN data."*

*"can be attributed to corona-type discharges occurring at close metallic objects near the receiving antenna in high local electric fields below the thundercloud."*
Can you localize where these discharges occurred? Do you have any report from staff seeing the corona discharges? Is it possible?
The observatory staff did not report any sparkling, but we did not expect them to see anything, as the thunderstorm occurred during the day. We added a sentence on line 520:

*"Note that visible sparks were not expected to be reported by the observatory staff during the daytime."*

*"a presence of strong corona discharges which might have been contributing to the delivery of additional positive charge to the cloud base. "*
This conclusion comes from the "the unipolar character of pulses "only?

This hypothesis is based on the pulse properties (duration, unipolar character of pulses, random time distribution, see above, Arcanjo et al., 2021) and on findings of Nag & Rakov (2009) and Chauzy & Soula (1999), who proposed the corona discharges as an additional hypothetical source of positive charge for the LPCR. We rephrased the statement on line 530, which now read as follows:

*"….c. a presence of irregularly distributed narrow unipolar pulses linked to strong corona discharges which might have been contributing to the delivery of additional positive charge to the cloud base.*

"LPCR inside the thundercloud is probably responsible for a high electric field in the bottom thundercloud dipole, which accelerated seed electrons and as a result, we observed significant long-lasting bremsstrahlung."
Why you emphasize the bottom of dipole and not the whole dipole between Main Negative layer and LPCR? Have you estimated the MN height from the radar data? The extension of strong electron accelerating field should be 1-2 km for TGE initiation. Please, give an estimate of the extent of the" high electric field" you mentioned. It will be good if you estimate roughly the size of the lower dipole between MN and LPCR.
By "*the bottom thundercloud dipole*", we meant the whole dipole formed by the LPCR and the main negative charge region, not only the bottom of the dipole. We changed the wording from "*the bottom thundercloud dipole*" to "*the lower thundercloud dipole between the LPCR and the main negative charge region*" on lines 32, 535 and 550.

Using our radar data, we are able to distinguish types of hydrometeors in the cloud and based on that we can suggest areas where cloud electrification occurred. Our radar is a cloud profiler (1D data) contrary to standard weather radars, which rotate (2D to 3D). The cloud profiler is not a fully polarimetric radar and does not measure the quantities KDP (Differential Reflectivity) or ZDR (Specific Differential Phase), which were used for example by Biggerstaff et al. (2017) together with the lightning mapping array data (which we don't have for our case) to estimate the charge distribution in the thundercloud.  We explain the limitation of the radar used in our study in the instrumentation section on lines 189-194:

*" Based on this classification we can suggest areas where cloud electrification occurred but our radar does not directly measure the charge structure of the cloud. It is not a fully polarimetric radar and does not measure the quantities like KDP (Differential Reflectivity) or ZDR (Specific Differential Phase), which were used, together with the lightning mapping array data, for example by Biggerstaff et al. (2017) to retrieve the locations of charge centres."*

To soften our statement we reworded the sentence (on lines 534-537) as follows:
*"LPCR inside the thundercloud is probably responsible for an adequately high electric field in the lower thundercloud dipole between LPCR and the main negative charge region extending over at least 2 km, as we can estimate it from the hydrometeor classes observed by the cloud radar. This extended charge structure was probably capable to accelerate seed electrons and, as a result, we observed significant long-lasting bremsstrahlung."*

We also added estimates of the published strengths of the in-cloud electric fields in the introduction section on lines 78-80:

*„Using the observed enhancements of photon and electron fluxes measured by the upper scintillator of SEVAN at Lomnicky stit (altitude 2 634 m) and their comparison with the simulations of the RREA Chillingarian et al. (2021) showed, that the potential difference present in in the thunderous atmosphere might reach approximately 500 MV."*

*"The meteorological situation allowed for a formation of a strong lower positive charge region located close to the observatory (only 180 m above it during the storm B). "*

Please, explain in more detail how you obtain the estimate of 180 m., please, give the height in limits, usually, LPCR extension is several hundred meters, and it is a transient phenomenon, changing and finally escaping with graupel fall.

As written above we assume that base of the cloud corresponds to the position of the lower edge of LPCR (Rakov & Uman, 2003) and that the bottom of the thundercloud corresponds to the lifted condensation level (Daidzic, 2019).  To express better the limitation of our measurements we reworded lines 546-548:

*"The meteorological situation allowed for a formation of a strong lower positive charge region with its lower edge located close to the observatory (only 180 m above it during the storm B assuming the lower edge of the LPCR is located at the cloud base)."*

**References**

1. Chilingarian, A., Hovsepyan, G., & Sargsyan, B. (2020). Circulation of Radon progeny in the terrestrial atmosphere during thunderstorms. *Geophysical Research Letters*, *47*, e2020GL091155. https://doi. org/10.1029/2020GL091155.

2. A. Chilingarian, G. Hovsepyan, E. Svechnikova, and M. Zazyan, Electrical structure of the thundercloud and operation of the electron accelerator inside it, Astroparticle Physics 132 (2021) 102615 https://doi.org/10.1016/j.astropartphys.2021.102615

3. Soghomonyan, Suren; Chilingarian, Ashot ; Khanikyants, Yeghia (2021), "Dataset for Thunderstorm Ground Enhancements terminated by lightning discharges", Mendeley Data, V1, doi: 10.17632/p25bb7jrfp.1

---

## Author Comment (AC2)

**We thank Martino Marisaldi for careful reading of the manuscript and for his very helpful comments and suggestions. We responded to all of them and revised the manuscript accordingly.** (The line numbers are related to the manuscript with tracked changes.)

**Responses to Reviewer (in blue) with Reviewer´s comments in black.**

1. Line 40: do the authors mean Kuroda et al. ? Kudela is not present in the reference list
Kudela et al. (2017) is the correct reference. By mistake, it was missing in the original reference list. This is fixed in the revised version.

*Kudela, K., Chum, J., Kollárik, M., Langer, R., Strhárský, I., & Baše, J.: Correlations between secondary cosmic ray rates and strong electric fields at Lomnický štít. Journal of Geophysical Research: Atmospheres, 122, 10,700–10,710. doi:10.1002/2016JD026439, 2017.*

2. Line 42: this term is widely used by the astrophysical community for gamma-ray bursts (GRB) of cosmic origin, and it can be misleading without additional explanation
We modified the wording on line 42 and used the same term "*gamma ray bursts of atmospheric origin*" as in the title of the paper by Brunetti et al. 2020.

3. Line 44: it is worth mentioning also the paper by Kelley et al., 2015 DOI: 10.1038/ncomms8845
We added the reference on line 44.

4. Line 69: and if the electric field is above the threshold for RREA
Added on line 70.

5. Line 80: regarding glow termination, I think that it is relevant the paper Kochkin et al., 2021 https://doi.org/10.1029/2020JD033467
We added the reference on line 90.

6. Line 80: again, is this Kuroda et al? This is not in the reference list
The reference to Kudela et al., 2017 was already added.

7. Line 137: is also the middle layer made of plastic scintillators? It is not clear from this description, but I think so. I would specify it.
The middle layer is also made of plastic scintillator. To make it more clear we add a specification on line 150:

"*and in the middle there is a thick 50 x 50 x 25 cm$^3$ scintillator stack (five standard plastic scintillator slabs).*"

And on line 153:
…" *the middle plastic scintillator stack of SEVAN,*"

8. Line 153: This means that 52 ms data before the trigger are collected. This is clarified at line 330 but it could be mentioned here for clarification.
We added the clarification on line 167:

*"…it records a 168-ms long waveform snapshot including a history of 52 ms before the trigger."*

9. Line 182: I guess the unit should be J/kg
Corrected on line 205.

10. Line 188: should this be m/s ?
Corrected on line 211.

11. Line 215: the estimate of the altitude of the cloud base is crucial. How reliable is the method?
We are aware of the fact, that the method of estimation of the cloud base from the LCL height is very rough. The error in LCL height estimated using this simple method could reach 15 % according to Lawrence (2005). The LCL height represents the altitude of the lowest possible cloud base. We added this information on lines 237-238.

*"The LCL height represents the altitude of the lowest possible cloud base and the error in the LCL height estimation when using this simple method could reach 15 % (Lawrence, 2005)."*

*Lawrence, M. G.: The Relationship between Relative Humidity and the Dew point Temperature in Moist Air: A Simple Conversion and Applications. Bulletin of the American Meteorological Society 86, 2, 225-234, doi:10.1175/BAMS-86-2-225, 2005.*

12. For storm A (Fig. 2) the LCL goes from about 1 km down to 400 m during the TGE, so assuming 400 m might not be the best choice, considering that the most intense part of the TGE happens when the LCL is higher than this value. This comment is also relevant to the conclusions section.
In the description of Fig. 2 we are now rather using the interval of altitudes, the wording is as follows on lines 240-242:

*"The altitude of the cloud base was estimated to decrease from 1100 to 200 m during storm A. During storm B, the height of the cloud base varied between 180 and 240 m.*

We also added following sentence on lines 292-293:

*"The most intense parts of the TGE events happened when the cloud base was located at about 800 m during storm A and at about 200 m during storm B."*

The wording in the conclusion section was modified as follows on lines 572-578:

*"The meteorological situation allowed for a formation of a strong lower positive charge region with its lower edge located close to the observatory assuming the lower edge of the LPCR was located at the cloud base at the beginning of the storm (Rakov and Uman, 2003). The altitude of the cloud base varied between 1100 and 200 m during the storm A and between 240 and 180 m during the storm B. Nevertheless, the LPCR is a transient phenomenon, which is moving down with positively charged falling graupels. Therefore, it is*

*probable, that the LPCR might have been located even closer to the detector during the graupel fall, when we observed the particle flux maxima.*"

13. Line 227: This comment is related to that at line 215. Here the authors also give the altitude of the cloud base, but now they give a range (correctly, in my opinion), while at line 215 it is give a single value. I think the two statements must be consistent.
We made both statements consistent, see our previous reply.

14. Line 315: I am not familiar with the EUCLID network. How do you think it is reliable the automatic classification of IC / CG of the network?
The evaluation of EUCLID IC/CG classification performance using the ground-truth data from high-speed camera records gave an average IC/CG discrimination accuracy of 80%. The misclassified discharges were the weakest ones. We add this information on lines 366-368:

"*Note that the IC/CG classification accuracy depends on the polarity and strength of the discharge and reaches on average 80%, while the misclassified strokes were generally very weak (Schwalt et al., 2020).*"

*Schwalt, L., Pack, S., and Schulz, W.: Ground truth data of atmospheric discharges in correlation with LLS detections, Electric Power Systems Research, Volume 180, 106065, doi: 10.1016/j.epsr.2019.106065, 2020.*

15. Line 325: Does this mean that many transients were not stored because of the maximum throughput of the acquisition system? If yes, it can be explicitly stated in the text.
Yes, we reached the limitations of the throughput of our acquisition system during the investigated time interval, when the signal level repeatedly and in a fast succession exceeded the triggering threshold. The modified sentence reads as follows on lines 372-374:

"*The maximum amount of 3-4 snapshots per second was recorded during the time of the intense variations of the atmospheric electric field, when the limitations of the throughput of our acquisition system were reached.*"

16. Line 332: since the signal is often saturated, I think that the numerically integrated value (Fig. 4a) might be inaccurate in many cases. If yes, I think it could be worth mentioning.
We noted it in the caption of figure 4:

"*Note that the numerically integrated values might be inaccurate because of a frequent saturation of the received signal.*"

17. Line 362: I was curious about this 2km boundary looking at Fig. 2 and 3. So, is this a real feature, and not some artifact? Honestly I was puzzled by the sharpness of the 2 km feature in both Z and vertical wind velocity.
The boundary visible at an altitude of about 2 km in the radar reflectivity plot (Fig. 2d) and in the vertical velocity plot (Fig. 2e) is not an artefact; it represents the impact of the melting layer on measurements of these quantities because of the sudden change in the hydrometeor composition at temperatures below and above 0°C.

We added following explanation on lines 184-186 and corresponding reference in the reference list.

*"The relatively narrow melting layer can be often detected in the radar reflectivity as a region with enhanced reflectivity, due to sudden changes in the hydrometeor properties (shape, size, and melting fraction) at temperatures below and above 0°C (Ryzhkov and Zrnic, 2019)."*

*Ryzhkov, A.V. and Zrnic, D.S.: Radar Polarimetry for Weather Observations, Springer, Berlin/Heidelberg, Germany, Volume 486, 2019.*

18. Line 389: In Fig. 3f I am puzzled by the CR boundary at a constant height of 2 km, as for storm A, while the Z profile shows a much larger variability. Can you briefly explain the origin of this feature according to the Sokol classification scheme?

The classification scheme uses the information about the altitude of the melting layer. Below the melting layer, snow or ice cannot exist because they have small terminal velocities and almost immediately melt in the melting layer or just below the melting layer. Therefore, only graupel, hail, cloud droplet and rain can appear between ground level and melting layer.

In case of the storm A, the altitude of the melting level can be derived from the radar reflectivity and vertical air velocity plots (Figs. 2d and 2e). In case of the storm B, where the signatures of the presence of melting lever are not visible in the radar reflectivity plot, we calculated its location using the temperature measurements conducted at the observatory.

We added this explanation in the radar description on lines 192-195:

*"The classification scheme uses the information about the altitude of the melting layer. Below the melting layer, snow or ice cannot exist because they have small terminal velocities and almost immediately melt in the melting layer or just below it. Therefore, only graupel, hail, cloud droplet and rain can appear between the ground level and the melting layer."*

We also added a clarification in the description of the storm A on lines 412-413 :

*"This altitude of the melting layer corresponded to the value calculated from the measured ground temperature (using a gradient of -6.5 C/km) and served as an input for the hydrometeor classification."*

A clarification for the storm B was added on lines 432-433:

*"The melting height is hardly definable from the radar measurements in this case, so for the hydrometeor classification we have to calculate it from the measured ground temperature."*

19. Line 394: This paper is not present in the reference list. Searching on Web of Science I found

Chilingarian, Vanyan and Mailyan (2013)
http://dx.doi.org/10.1016/j.astropartphys.2013.06.006 which might be the right paper.
Please check the citation.

The paper cited in the original version (Chilingarian and Mailyan, 2013) was a conference paper which was mistakenly omitted in the reference list. As we found the same energy spectra in a paper in Atmospheric Research from 2012, we corrected the citation to "*Chilingarian et al., 2012*" on line 445. This reference was already present in the reference list.

*Chilingarian, A., Mailyan, B., and Vanyan, L.: Recovering of the energy spectra of electrons and gamma rays coming from the thunderclouds, Atmospheric Research 114–115, doi:0.1016/j.atmosres.2012.05.008, 2012.*

20. Line 408: It is not clear to me if this number takes into account the detector and environment geometry just described or not.

The number takes into account the detector and environment geometry. We modified the wording accordingly on lines 459:

"*The PHITS's cosmic rays source for the specific date, height, and geometry gives us a total number of….*"

21. Line 425: If I understand correctly, only the count rate above the 6.5 MeV threshold is measured, and not the energy of every single count. I think this should be stated at an earlier stage in the manuscript, when the particle detector setup is described.

The energy of individual detected particles is not evaluated. We added this statement to the SEVAN description in the Instrumentation section on line 157.

22. Line 427: This is using the spectrum from Chilingarian and Mailyan 2013, if I understand correctly, and not the real measurements. I think this should be stated explicitly also here in the caption.

We used the energy spectra measured by Chilingarian et al., 2012 (their Fig. 12) as an input of our model and for calculations of the deposit energy spectra in the modelled detector setup. We modified the caption of Fig. 5, which now reads:

"*Figure 5: a) TGE source particles, the source is tilted by 45° from the vertical axis, and the energy spectra correspond to observation by Chilingarian and Mailyan et al. (2012), their Fig. 12. b) Calculated deposit energy spectra inside the scintillator. Lower energy threshold was set to 6.5 MeV according to the setup of the detector. The detector itself does not provide the energy spectra.*"

23. Line 428: The enhancement over the background is similar, but what is the background level for that event at Aragats? Is it consistent with the one in this paper?

The background count level registered at Aragats by the middle SEVAN scintillator before the event from 4th October 2010 was about 7100 per minute (http://adei.crd.yerphi.am/). To compare the TGE events at Milešovka and at Aragats at very different altitude we rather changed the relative enhancement to its absolute value. The statement was modified as follows on lines 479-481:

*"…when the count enhancement in the middle SEVAN scintillator reached about 1400 counts/min, which is similar to our observation. The background level was about 7100 counts/min."*

24. Line 433: Provided that Chilingarian, Vanyan and Mailyan (2013) is the right paper to cite (see comment at line 394) then the spectral parameters for the 4 Oct 2010 TGE included in Fig. 10 of that paper are different from those used here. The constants are 3.5x10^5 for 5-10 MeV range, and 4.2x10^7 for the 10-100 MeV range. Which parameters have been really used? This would impact the expected count rate reported later at lines 440-441

The constants for the same event are unfortunately different in the papers by Chilingarian, Vanyan and Mailyan (2013) and Chilingarian, Vanyan and Mailyan (2012).
We have used those from the 2012 paper, their Fig. 12. We corrected the citation in the manuscript on line 487.

25. Line 439: I am not sure the comparison shown here is very significant, for various reasons:
- what was the background level at Aragats for this reference measurement? (see comment at line 428) - looking at Figure 5a it seems that the total number of counts is very sensitive to the inclination of the beam. Just a little larger inclination and you would get the unabsorbed beam directly onto the detector, sending the count-rate sky high.
Therefore I think that this comparison is very dependent on the assumption on geometry, which cannot really be validated. If the authors want to keep this part, I recommend to comment on the above points explicitly in the text.

For the background level, please see our reply to comment 23. We admit that the comparison is very dependent on the geometry assumption. Therefore, we removed the comparison from the simulation section and added following statement on lines 498-501.

*"We have calculated the count enhancements also for other inclinations of the beam. We verified that the enhancements calculated for an inclination of 45° best reproduced our measurements. To obtain the same values for a beam arriving more vertically to the detector we would need to assume a stronger TGE. a more horizontal inclination of the beam seems to be not realistic."*

26. Line 456: This statement is affected by my comments at line 439
We softened substantially the statement, which now reads as follows on lines 512-513:

*"Using the simulations, we have shown that the observed increases of count rates might have been related to TGEs."*

27. Line 460: This is a very interesting discussion point. I understand that the station is one of the stormiest places in the Czech republic, so it would be useful if the authors could provide some figure of merit for this, for example the number of stormy days, or a similar indicator if available.

We added a reference to a recent paper by Novak and Kyznarova (2020) and modified the wording on line 144-5 as follows:

*"It is located in the stormiest region in the Czech territory with about 3.2 CG flashes/km$^2$/year (Novak and Kyznarova, 2020; Fig. 9a).*

*Novak, P. and Kyznarova, H.: Long-term characteristics of convective storms in terms of radar data and lightning detection data, Meteorological bulletin, Vol. 73,  ISSN 0026-1173, 2020.*

[Figure]

**Obr. 9 Průměrný roční počet blesků do země za období 2002–2016. Velikost gridového bodu je 1 × 1 km (a) a 20 × 20 km (b).**
Fig. 9. The annual average of cloud to ground flashes for 2002–2016 period. Grid point size is 1 × 1 km (a) and 20 × 20 km (b).

28. Line 465: According to Fig. 2a the cloud base, as parameterized by the LCL, is higher during the first part of the TGE, which is also the most intense. If this is true, this points also to a larger intensity in the first part of the event.

The wording in the conclusion section was modified as follows on lines 573-578:

*"The meteorological situation allowed for a formation of a strong lower positive charge region with its lower edge located close to the observatory assuming the lower edge of the LPCR was located at the cloud base at the beginning of the storm (Rakov and Uman, 2003). The altitude of the cloud base varied between 1100 and 200 m above the observatory during the storm A and between 240 and 180 m during the storm B. Nevertheless, the LPCR is a transient phenomenon, which is moving down with positively charged falling graupels. Therefore, it is probable, that the LPCR might have been located even closer to the detector during the graupel fall, when we observed the particle flux maxima."*

29. Line 477: This is an important statement but I did not find it substantiated in the text
We removed the misleading part, which described the usual observation of near surface electric field observed during TGEs at high altitude observatories, and we limited ourselves to the summary of our unusual observation.

30. Line 482: The paper by Kochkin et al, 2021, already mentioned at line 80, shows cases where a glow flux is abruptly reduced, but not terminated, by lightning discharges.
We add the finding by Kochkin et al. 2021 on line 539:

*"….none of which, however, abruptly reduced or terminated the TGE flux."*

And on line 88-90:

*"….but quite often they are reduced or terminated abruptly by a nearby lightning discharge (Kudela et al., 2016; Chilingarian et al., 2017a; Chum et al., 2020, Soghomonyan et al., 2021; Kochkin et al., 2021)."*

31. Line 484: Also this statement, which is very relevant, I think is not sufficiently addressed in the main text. At least, it would be important to show in details one case where this radiation increase occurs. Considering that radiation is recorded in minute-long time bins, I think it is difficult to get a conclusive correlation with electric field data. This point was also raised by the previous reviewer.
We addressed this issue by adding a sentence on lines 539- 541:

*"We cannot exclude that short duration TGE events of a few tens of seconds could have been reduced (Kochkin et al, 2021) or terminated (Chum et al., 2020; Chilingarian, 2017b, 2020) by a lightning stroke, as these would not be recognisable in the 1-min cadence SEVAN data."*

32. Line 493: can this be the peculiar characteristic of these storms, which is missing in the majority of the storms passing over the observing site? A suggestion for a follow-up study could be to test the presence of a large LPCR in other storms without recorded TGE.
We added as suggested at the end of discussion section on line 581:

*"A follow-up study is needed to test the absence of a large LPCR in other storms without recorded TGEs."*

---

## Author Response (AR2)

**We thank Martino Marisaldi for careful reading of the manuscript and for spotting the missing reference.**

We added the reference to Ryzhkov and Zrnic (2019) into the reference list.

*Ryzhkov, A.V. and Zrnic, D.S.: Radar Polarimetry for Weather Observations, Springer, Berlin/Heidelberg, Germany, Volume 486, 2019.*